



# 1 A synthesized field survey database of vegetation and active
# 2 layer properties for the Alaskan tundra (1972-2020)

Xiaoran Zhu[1], Dong Chen[2], Maruko Kogure[2], Elizabeth Hoy[3,4], Logan T. Berner[5], Amy L.
Breen[6], Abhishek Chatterjee[7], Scott J. Davidson[8,9], Gerald V. Frost[10], Teresa N.
Hollingsworth[11,12], Go Iwahana[6], Randi R. Jandt[6], Anja N. Kade[13], Tatiana V. Loboda[2], Matt J.
Macander[10], Michelle Mack[14], Charles E. Miller[7], Eric A. Miller[15], Susan M. Natali[16], Martha K.
Raynolds[17], Adrian V. Rocha[18], Shiro Tsuyuzaki[19], Craig E. Tweedie[20], Donald A. Walker[17],
Mathew Williams[21], Xin Xu[2], Yingtong Zhang[1], Nancy French[22], Scott Goetz[5]
[1]Department of Earth & Environment, Boston University, Boston, Massachusetts, 02215, USA
[2]Department of Geographical Sciences, University of Maryland, College Park, Maryland, 20742, USA
[3]NASA Goddard Space Flight Center, Greenbelt, Maryland 20771, USA
[4]Global Science & Technology, Inc., Greenbelt, Maryland 20770, USA
[5]School of Informatics, Computing, and Cyber Systems, Northern Arizona University, Flagstaff, Arizona 86004,
USA
[6]International Arctic Research Center, University of Alaska Fairbanks, Fairbanks, Alaska 99775, USA
[7]Jet Propulsion Laboratory, California Institute of Technology, Pasadena, California 91109, USA
[8]School of Geography, Earth and Environmental Sciences, University of Plymouth, Plymouth, PL3 4PA, UK
[9]Department of Geography and Environmental Management, University of Waterloo, Waterloo, N2L 3G1, Canada
[10]Alaska Biological Research, Inc., Fairbanks, Alaska 99775, USA
[11]Aldo Leopold Wilderness Research Institute, Rocky Mountain Research Station, Missoula, Montana 59801, USA
[12]Boreal Ecology Team, PNW Research Station, Fairbanks, Alaska 99775, USA
[13]Department of Biology and Wildlife, University of Alaska Fairbanks, Fairbanks, Alaska 99775, USA
[14]Department of Biological Sciences, Northern Arizona University, Flagstaff, Arizona 86004, USA
[15]Bureau of Land Management Alaska Fire Service, Fort Wainwright, Alaska 99703, USA
[16]Woodwell Climate Research Center, Falmouth, Massachusetts 02540, USA
[17]Institute of Arctic Biology, University of Alaska Fairbanks, Fairbanks Alaska 99775, USA
[18]Department of Biological Sciences, University of Notre Dame, Notre Dame, Indiana 46556, USA
[19]Graduate School of Environmental Earth Science, Hokkaido University, Sapporo, 060-0810, Japan
[20]Department of Biological Sciences and the Environmental Science and Engineering Program, The University of
Texas at El Paso, El Paso, Texas 79968, USA
[21]School of GeoSciences, University of Edinburgh, Edinburgh, EH9 3FF, UK
[22]Michigan Tech Research Institute, Michigan Technological University, Ann Arbor, Michigan 48105, USA

*Correspondence to*: Dong Chen (itscd@umd.edu)





**Abstract.** Studies in recent decades show strong evidence of physical and biological changes in the Arctic tundra largely in response to exceptionally rapid rates of warming. Given the important implications of these changes on ecosystem services, hydrology, surface energy balance, carbon budgets, and climate feedbacks, research on the trends and patterns of these changes is becoming increasingly important and can help better constrain estimates of local, regional, and global impacts as well as inform mitigation and adaptation strategies. Despite this high need, scientific understanding of tundra ecology and change remains limited largely due to the inaccessibility of this region and less intensive study compared to other terrestrial biomes. A synthesis of existing datasets from past field studies can make field data more accessible and open up possibilities for collaborative research as well as for investigating and informing future studies. Here, we synthesize field datasets of vegetation, and active layer properties from the Alaskan tundra, one of the most well-studied tundra regions. Given the potential increasingly intensive fire regimes in the tundra, fire history and severity attributes have been added to data points where available. The resulting database is a resource that future investigators can employ to analyze spatial and temporal patterns in soil, vegetation, and fire disturbance-related environmental variables across the Alaskan tundra. This database, titled Synthesized Alaskan Tundra Field Database (SATFiD), can be accessed at the Oak Ridge National Laboratory Distributed Active Archive Center (ORNL DAAC) for Biogeochemical Dynamics (Chen et al., 2023: https://doi.org/10.3334/ORNLDAAC/2177).

## 1 Introduction

Over recent decades, the Arctic tundra has warmed three to four times faster than the global average (Rantanen et al., 2022), leading to profound physical and biological changes. Over this period, shrubs and trees have become more abundant in both the North American and Eurasian Low Arctic (Hagedorn et al., 2014; Rees et al., 2020; Mekonnen et al., 2021; Dial et al., 2022). Across the Arctic tundra, as defined by the circumpolar Arctic bioclimatic subzones map (CAVM Team, 2003; Walker et al., 2005; Raynolds et al., 2019), a lengthening of the growing season has been observed due to rising temperatures (Goetz et al., 2005; Ernakovich et al., 2014; Arndt et al., 2019). At the same time, widespread increases in vegetation productivity have been documented by both field measurements (Myers-Smith et al., 2020) and satellite observations (Goetz et al., 2005; Berner et al., 2020). While the direct mechanisms underlying Arctic "greening" are complicated and vary among ecosystems (Rocha et al., 2018; Myers-Smith et al., 2020), it is believed these mechanisms are fundamentally driven by the increasingly favorable growing conditions for vegetation created by warming, including longer growing seasons (Goetz et al., 2005; Arndt et al., 2019; Berner et al., 2020). Moreover, because of this warming, carbon-rich permafrost across the Arctic tundra has shown signs of thawing (Lewkowicz and Way, 2019; Heijmans et al., 2022). Permafrost degradation is apparent through the increasing occurrence of thermokarst and deepening of the active layer thickness (ALT), both of which have contributed to increased nutrient availability and a changing cover of surface water bodies across the Arctic tundra (Schuur et al., 2007; Chen et al., 2021). Additionally, wildfires, while historically rare during recent geological periods, are a significant disturbance agent that may have entered a stage of increasing severity, frequency, and extent (French et al., 2015; Hu et al., 2010). Altogether, these physical and biological changes have



profound implications for the global carbon cycle, energy budget, land-atmosphere interactions, and future state of
the tundra (Oechel et al., 1993; Chapin et al., 2005; Mack et al., 2011; Schuur et al., 2015).
Considering the Arctic tundra's important role in the Earth system and the strong warming in this region,
understanding current ecosystem dynamics is crucial for the projection of future states of the Arctic tundra.
Additionally important is understanding the subsequent changes in ecosystem services and land-atmosphere
interactions occurring in a changing Arctic. Despite the vast expanse of Arctic tundra and its high susceptibility to
sustained warming, our collective understanding of the ecological processes that occur within the tundra remains
limited. This historical lack of studies compared with other biomes is the consequence of limited *in situ*
measurements, stemming from interwoven factors including harsh Arctic environmental conditions, logistical
challenges, and the high cost of conducting scientific expeditions.
The Alaskan tundra is an important component of the Arctic tundra biome that spans over 8.5 million $km^2$ and
makes up slightly more than 7% of the total circumpolar Arctic area (CAVM Team, 2003). It is one of the few
wildfire "hotspots" across the circumpolar tundra in recent decades (Masrur et al., 2018). Thanks to efforts by state
and federal fire management agencies, the Alaskan tundra has one of the longest and highest quality wildfire records
of any Arctic region, with the earliest spatially-explicit wildfire record dating back to the early 1950s. However,
even these early records of wildfires across the region are sparse, and often only larger wildfires were included,
leading to unaccounted wildfires in the region (Miller et al., 2023). Additionally, the Alaskan tundra is arguably one
of the most studied tundra regions in the world. To our knowledge, field measurements of vegetation and active
layer properties conducted in the Alaskan tundra were mentioned in the literature as early as 1889, and the USGS
began field surveys of geography and geology in 1889 (Schrader, 1902; Russell, 1890). Moreover, dedicated field
stations such as the Toolik Field Station (est. 1975), a part of the Arctic Long Term Ecological Research Network
(LTER), and the Barrow Arctic Research Center/Environmental Observatory (est. 1973) have greatly facilitated
scientific discovery in the region.
Despite the fact that many in situ datasets recorded in the Arctic tundra have been made publicly available, they are
scattered across data repositories. Additionally, it is not uncommon for field datasets to be referenced in published
literature while the datasets themselves were never publicly released. While all existing field datasets are important
in their own right (in support of the scientific goals of the individual field campaigns), we argue that when combined
properly they can provide an unprecedented lens through which the ecosystem dynamics of the Arctic tundra, both
aboveground and below-ground, can be revealed at a wide spatial scale. To our knowledge, there has not been an
effort to compile field datasets on vegetation, active layer properties, and fire attributes, collected in different parts
of the Alaskan tundra and reconciled into a consistent database. Because of this, we built a database from *in situ*
datasets across the Alaskan tundra with three major objectives: (1) Gather datasets and synthesize them in a way that
will facilitate further analysis by investigators and promote synthesis research efforts, (2) deepen our understanding
of ecosystem processes within the Alaskan tundra, particularly fire-vegetation-permafrost interactions, and (3)



identify areas of interest for future research where knowledge is lacking or there is great potential for follow-up
research to study change and long-term trends.
**Study Area**
This database, titled Synthesized Alaskan Tundra Field Database (SATFiD), synthesizes field-based datasets from
the Alaskan tundra as defined by the Circumpolar Arctic Vegetation Map (CAVM) (CAVM Team, 2003; Walker et
al. 2005; Raynolds et al. 2019). Data from this area can be further categorized by four major subregions: the North
Slope, Noatak, Seward Peninsula, and Southwest Alaska (Fig. 1). These subregions span a large range of climatic
and topographic conditions. In the North Slope, the northernmost Arctic Coastal Plain ecoregion is located in
Bioclimate Subzone D of the Circumpolar Arctic Vegetation Map and is characterized by flat, poorly-drained
lowlands with herbaceous and dwarf-shrub vegetation and a mosaic of water bodies (CAVM Team, 2003; Gallant et
al., 1995). All Alaskan tundra south of the Arctic Coastal Plain ecoregion lie within Subzone E of CAVM and is
generally warmer and more densely vegetated (CAVM Team, 2003). Within this subzone, farther inland in the
North Slope, is the Arctic Foothills ecoregion, which experiences warmer summer temperatures and features rolling
hills, more distinct drainage networks, and taller, extensive shrub cover (Gallant et al., 1995). The Noatak subregion
follows the Noatak River Valley and has a dry climate compared to the Seward Peninsula to its south (He et al.,
2021). The Southwest is the warmest subregion of the Alaskan tundra. It consists of coastal plains with wet soils and
shallow active layers, and winding rivers and streams (Gallant et al., 1995).
**3 Data and methods**
**3.1 Data**
Datasets compiled into SATFiD were obtained from three main sources: (1) direct correspondence with principal
investigators, (2) data repositories including the Oak Ridge National Laboratory Distributed Active Archive Center
(ORNL DAAC) and the Environmental Data Initiative (EDI), and (3) a systematic search for literature that was
based on field data collected in the Alaskan tundra. Permission was obtained from each principal investigator for
incorporation of their datasets in this synthesis. A list of these original datasets and access to ones that are published
and publicly available are included in Appendix A (Table A1). These datasets spanned many research projects with
diverse research foci pertaining to the Alaskan tundra. That translates to specific variables included in the original
datasets that vary greatly. Even for the same variables, sampling frequency, and number of samples,
instrumentation, and methodology often varied by project. To create a database that can advance capacity for
synthesis research on the Alaskan tundra, variables were selected for inclusion in the database (section 3.2) and
these data were standardized and filtered (section 3.3).
The individual datasets that were ingested defined plots that varied in size, sampling within sites versus along
transects, and sampling techniques. For consistency, we define unique data points as points that were collected at
unique latitude, longitude, and collection dates as provided in the original datasets.
**3.2 In-situ variables selection**



The variables included in SATFiD (shown in Table 1) were selected from the incorporated datasets with a goal of
preserving variables that were gathered frequently in the various studies and are most relevant to the study of
Alaskan tundra vegetation and active layer properties. In addition to the field data variables, data descriptors and
wildfire-related variables were added to our database. The data descriptors include the assigned plot ID, dataset ID,
dataset name, latitude, longitude, date of collection, and year of collection. For each data point, the dataset ID and
name link it to its original dataset. These variables were added to facilitate the use of our database and also to allow
the users to be able to trace back the original datasets when such a need arises. The geospatial and remote-sensing
based wildfire-related variables were added to link data points to the known wildfire history at each point (since
wildfire plays a critical role affecting the aboveground and belowground conditions of tundra ecosystems). In total,
34 variables are contained by SATFiD (Table 1). Ground-based burn severity variables are not included in this
database as their collection methods were inconsistent across datasets, including various qualitative or quantitative
measures of severity that could not be reconciled into a single variable.
**Table 1 List of data variables included in SATFiD. Fire history attributes are sampled from the Alaska Large Fire**
**Database (ALFD) (Alaska Large Fire Database | FRAMES, 2022), and dNBR is sampled from the Landsat-derived Burn**
**Scar dNBR dataset (Loboda et al., 2018).**

| Field | Description |
| --- | --- |
| PLOT_ID | A unique ID for every plot included |
| DATASET_ID | Dataset ID number |
| DATASET_NAME | Name of dataset |
| LATITUDE | Latitude of plot |
| LONGITUDE | Longitude of plot |
| DATE | Date of data collection (YYYYMMDD) |
| PLOT_ORIGINAL_ID | Plot ID as defined in original dataset |
| SOIL_TEMP_10CM_C | Temperature at 10 cm depth (ºC) |
| PH | Soil pH |
| WATER_TABLE_CM | Water table (cm) |
| SOIL_MOIST_% | Volumetric water content (%) |
| ALT_MEAN_CM | Active layer thickness (cm) |
| ORG_SOIL_DEPTH_CM | Organic soil depth (cm) |
| LAI_MEAN | Leaf area index |
| SHRUB_HEIGHT_CM | Shrub height (cm) |
| STEM_COUNT | Shrub stem count per square-meter |



| | |
|---|---|
| MOSS_COVER_% | Moss cover (%) |
| LICHEN_COVER_% | Lichen cover (%) |
| GRAMINOID_COVER_% | Graminoid cover (%) |
| FORB_COVER_% | Forb cover (%) |
| SHRUB_COVER_% | Shrub cover (%) |
| BARE_COVER_% | Bare soil cover (%) |
| LITTER_COVER_% | Litter cover (%) |
| HARV_BIO_G/M^2 | Harvested aboveground biomass, oven-dried (g/m^2) |
| YR_DATA | Year of data collection (YYYY) |
| BURNED_STATUS | Whether or not plot was burned in the past at the time of data collection |
| FREQ_PRE | Number of times wildfires occurred prior to data collection |
| YR_LFIRE | Year of last known wildfire before data collection |
| N_YR_LFIRE | Number of years between last known wildfire before data collection and data collection |
| DNBR | dNBR of the last known wildfire before data collection |
| ALL_FIRE_YRS | Years of all known wildfires occurred at this point (comma-separated) |
| YR_NFIRE | Year of next known wildfire after data collection |
| N_YR_NFIRE | Number of years between data collection and next known wildfire after data collection |
| FREQ_TOTAL | Number of times wildfires occurred based on known wildfire history |


### 3.3 Data standardization and cleaning

Multiple types of data standardization were implemented to reconcile the ingested datasets. These standardization
decisions are listed in Table 2.
**Table 2: List of basic data standardization procedures.**

| Procedure | Description |
|---|---|
| Clipping | Because original datasets came from studies with varying study areas and ecosystems, data points from each dataset were initially clipped to only include points within the Alaskan tundra study area (with the exceptions being the plots that were confirmed by the original data collectors to be located in tundra), whose boundary is adopted from CAVM (Walker et al., 2005; CAVM Team, 2003). |



| | |
|---|---|
| Coordinate unification | The coordinates of the plots that were not in World Geodetic System 84 (WGS 84) were converted to WGS 84 decimal degrees. |
| Date conversion | All date values were converted into "YYYYMMDD" format. If a data point's collection month and/or day were unrecorded, their values were set to 0. |
| Data filtering | When multiple versions of the same variable existed in the original dataset, the version that was most similar to the same variable in the majority of datasets was kept. Examples of such situations include soil temperature (measurements at different depths were conducted by several datasets) and vegetation cover (Dataset Frost_2020 contains three types of vegetation cover: top-hit cover, any-hit cover, and multi-hit cover. Among these we picked the top-hit cover). |
| Unit unification | Required calculations were conducted to convert different units when they are used by different datasets. For example, soil moisture in terms of volumetric water content was calculated for Dataset Shaver_2016 by multiplying the provided gravimetric water content by bulk density. |
| Vegetation cover unification | In our database, vegetation cover is provided for main Plant Functional Types (PFTs), including shrub, moss, lichen, graminoid, forb, and litter. When only species-based vegetation cover was provided by a given dataset, we calculated the vegetation cover value of a given PFT by summing up all vegetation cover values of the individual species belonging to that PFT. |
| Daily mean calculation | Repeat measurements taken from a single plot, as defined by the latitude and longitude, within a given day were averaged for all quantitative variables. |


### 3.4 Fire history and severity sampling

### 3.4.1 Sampling fire history data from the Alaska Large Fire Database (1940-2021)

The Alaska Large Fire Database (ALFD) is the longest and most comprehensive spatially-explicit record of fire history in Alaska. Particularly for the tundra, where fire is historically scarce, the ALFD is useful for capturing relatively small fire scars compared to the larger scars found in the neighboring boreal forests, making it a useful tool for identifying fire history at a fine spatial scale. Fires in the ALFD are defined as fires at least 1,000 acres in area, but spatial resolution improves dramatically through the record, with fires of down to 10 acres included by 2015. Please see the Uncertainty section (Section 5.2) for a more detailed breakdown of how the ALFD defines large fires and a discussion of implications.

We used the ALFD to sample fire history data to each individual data point. Eight fire-related variables were added by sampling fire history polygons that data points intersected. Approximately 17% of the data points in this database were sampled at locations that fall within ALFD fire perimeters (Fig. 3). If a point was within a fire polygon from before the data sampling date, the point was labeled "Burned" in the BURNED_STATUS field. FREQ_PRE is the total count of past fire polygons the data point intersects. YR_LFIRE is the year of the most recent fire prior to the data point being sampled. N_YR_LFIRE is the year of data collection minus the year of the most recent past fire. ALL_FIRE_YRS is a list of fire years for all fire polygons intersected by the data point. YR_NFIRE represents the



year of the most recent fire after the data point was sampled. N_YR_NFIRE is the year of the next fire minus the
year of data collection. FREQ_TOTAL is a count of years in ALL_FIRE_YRS, representing the total number of fire
polygons intersected by the data point. Our database currently extends to 2020 and samples fire history data from the
2021-updated version of ALFD, but several large tundra fires have occurred since then. These can be incorporated
along with additional field datasets in future versions of the database.
**3.4.2 Sampling fire severity data from the Landsat-derived Burn Scar dNBR dataset (1985-2015)**
A dNBR attribute was sampled to data points from the Landsat-derived Burn Scar dNBR dataset (Loboda et al.,
2018). Rasters covering the tundra region of the ABoVE domain were mosaiced for each unique fire year associated
with the data points. For each burned point, a dNBR value from the mosaicked raster was sampled if available. The
values were then filtered to remove values of -3000, which represents no data, and -2500, which indicates invalid
pixels due to factors such as cloud cover.
**4 Results**
**4.1 Database overview**
SATFiD synthesizes 197,830 individual data points gathered from across 37 datasets. The data span the North
Slope, Noatak, Seward Peninsula, and Southwest subregions of the Alaskan tundra. A large cluster of points can be
seen on the North Slope in the area of the 2007 Anaktuvuk River Fire scar, which is a notable study point for tundra
fire research, as well as the continuous north-south transect along the Dalton Highway. Seventeen clustered data
points in the Seward Peninsula subregion from Jandt_1995 fall outside of the CAVM definition of tundra. These are
data from the Bureau of Land Management (BLM) and have been confirmed as tundra points (Fig. 1).

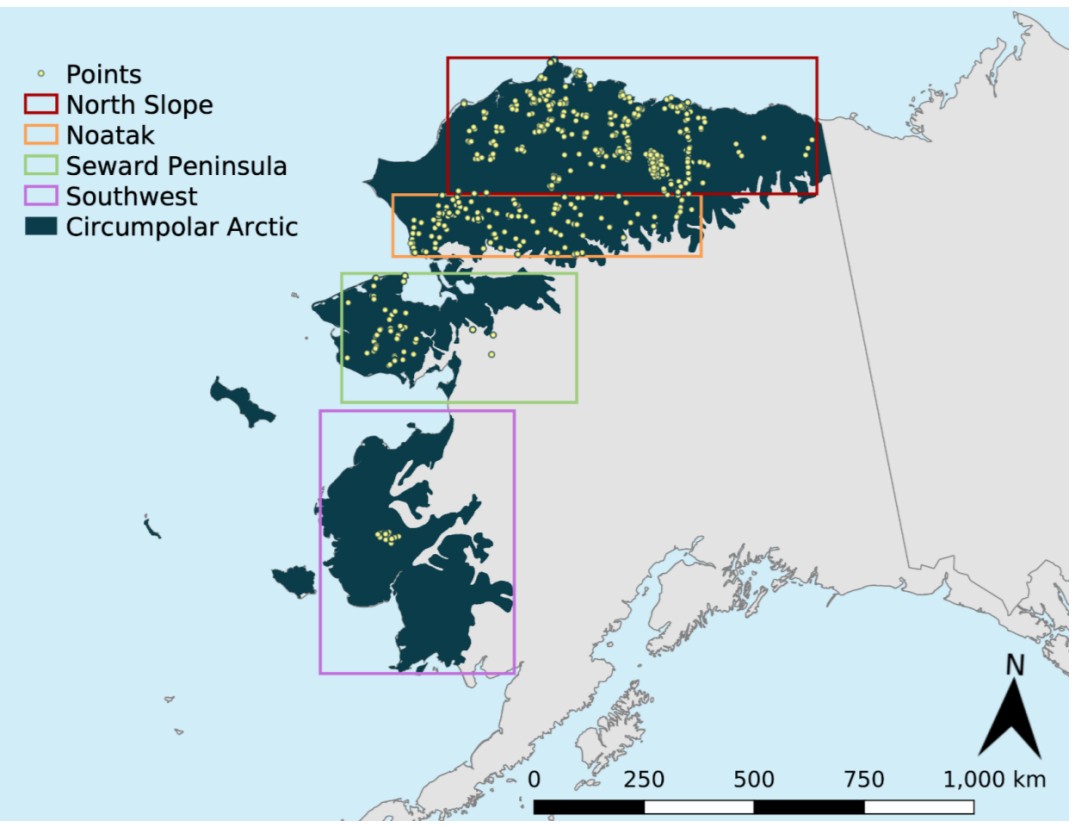


**Figure 1: Map of all points from 1940 through 2021 overtop the Circumpolar Arctic as defined in CAVM clipped to the**
**state of Alaska. 17 of the data points lie outside the CAVM definition of tundra. These points were sampled by BLM and**
**are tundra points. The colored reference boxes indicate the location of points within the circumpolar Arctic and are used**
**to define regions for this study.**
We note that each dataset has unique variables sampled and total number of data points. Many variables are
measured across multiple datasets, with the most frequently sampled variable across studies being shrub cover,
which can be found in 23 datasets. Second in greatest coverage across datasets are lichen cover and active layer
thickness, which appear in 22 datasets (Fig. 2, Table 3). The active layer thickness variable is dominated by the
Schaefer_2021 dataset, which is 192,483 data points, making up 98.6% of active layer thickness measurements and
97.3% of the data points in the database. It is very important to note, however, that despite the large quantity of data
points, the Schaefer_2021 dataset only includes measurements of active layer thickness and a relatively small
number of soil moisture measurements (4,892 points); hence, this dataset is not overrepresented in our synthesis and
in fact does not contribute to any other field-collected variable in this synthesis.



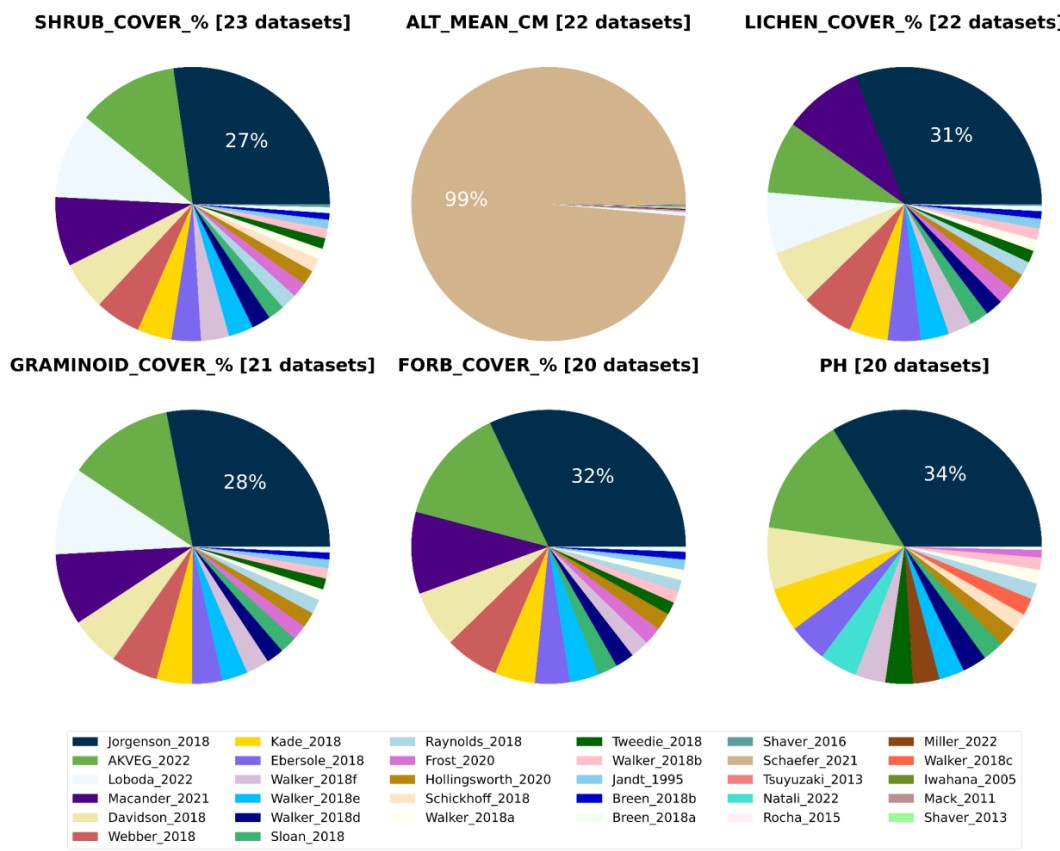


**Figure 2: Pie charts showing the distribution of how many data points each dataset contributes to the six field collected variables that appear the most across datasets. The top center pie chart indicates that the Schaefer_2021 dataset contributed overwhelmingly to active layer thickness data, but as the neighboring pie charts demonstrate, data for other variables are more evenly distributed across datasets.**

**Table 3: Field-based and fire-related variables by the number of datasets and data points they appear in.**

| Field type | Field | Number of datasets | Number of data points |
|---|---|---|---|
| Field Data | SOIL_TEMP_10CM_C | 6 | 2389 |
| | PH | 20 | 1915 |
| | WATER_TABLE_CM | 4 | 768 |
| | SOIL_MOIST_% | 10 | 6966 |
| | ALT_MEAN_CM | 22 | 195066 |
| | ORG_SOIL_DEPTH_CM | 15 | 1512 |
| | LAI_MEAN | 7 | 127 |



| | | | |
|---|---|---|---|
| | SHRUB_HEIGHT_CM | 13 | 865 |
| | STEM_COUNT | 2 | 197 |
| | MOSS_COVER_% | 13 | 1835 |
| | LICHEN_COVER_% | 22 | 2161 |
| | GRAMINOID_COVER_% | 21 | 2380 |
| | FORB_COVER_% | 20 | 2079 |
| | SHRUB_COVER_% | 23 | 2452 |
| | BARE_COVER_% | 17 | 1699 |
| | LITTER_COVER_% | 9 | 1216 |
| | HARV_BIO_G/M^2 | 5 | 222 |
| Fire Attributes | BURNED_STATUS | 37 | 197830 |
| | FREQ_PRE | 17 | 11070 |
| | YR_LFIRE | 16 | 10902 |
| | N_YR_LFIRE | 16 | 10902 |
| | DNBR* | 12 | 5567 |
| | ALL_FIRE_YRS | 37 | 58503 |
| | YR_NFIRE | 10 | 22871 |
| | N_YR_NFIRE | 10 | 22871 |
| | FREQ_TOTAL | 37 | 197830 |

*Extracted from intersected 30 m pixels in the Landsat-derived Burn Scar dNBR dataset (Loboda et al., 2018)
**4.2 Descriptive analysis of data by fire attributes**
Fire history information from the ALFD allows for the database to be grouped by whether and when points fell
within fire perimeters. If a point in a fire perimeter was sampled after the fire, it can be labeled "post-fire", and if the
point was sampled before the fire, it can be labeled "pre-fire". In the following figures, we define points that are in
fire perimeters from years before and after sampling as "pre-fire" and "post-fire" respectively. Of course, analysis
through different grouping methods may be equally if not more interesting to pursue depending on the study of
interest. What we present here is one of many ways to explore the data.
83% of the data points, 164,118 data points total, came from points that did not have any fire history since 1940
according to the ALFD. These are considered "unburned" in recent, recorded fire history although they could have
been burned prior to 1940. Out of burned points, 10,847 data points were sampled post-fire and 22,865 were
sampled pre-fire (Fig. 3: (a)). A parallel plot showing the distribution after excluding the Schaefer_2021 dataset of
mostly active layer thickness measurements is presented for comparison (Fig. 3: (b)). Within this subset, points with
fire history make up 46% of the data points.

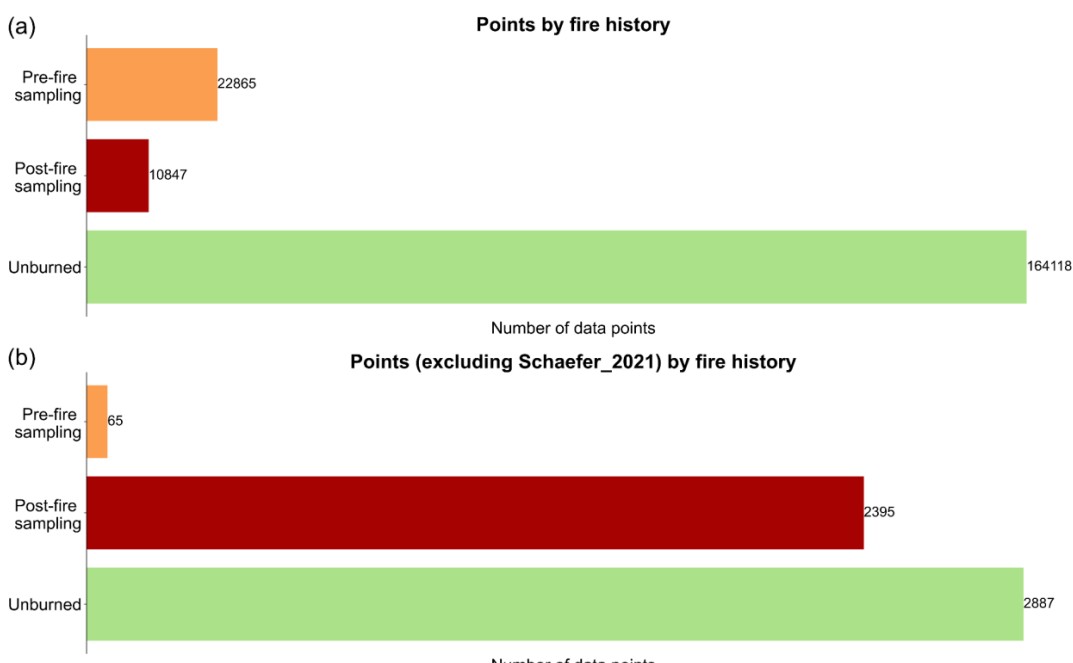


**Figure 3: (a) Data sorted by if and when the point was burned relative to sampling using fire perimeters from the ALFD,**

**(b) data excluding the Schaefer_2021 dataset by if and when the point was burned relative to sampling using fire**

**perimeters from the ALFD.**



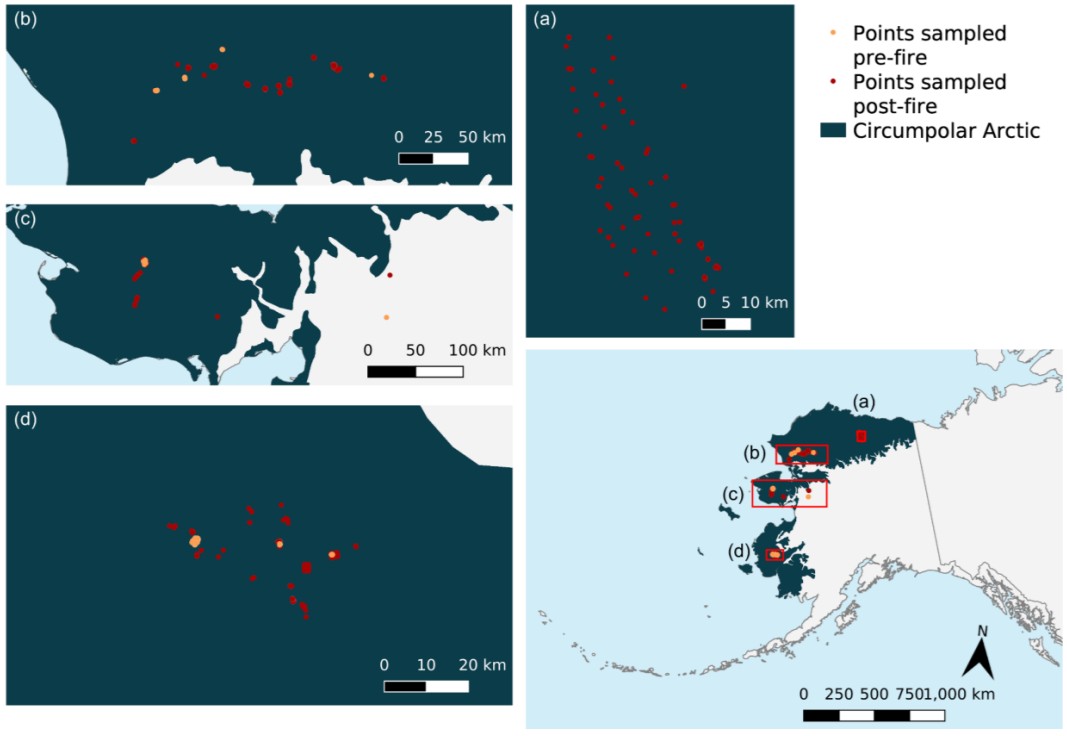


**Figure 4: Fire history for data points by subregion. Insets (a)-(d) show points with fire history in the (a) North Slope, (b)**
**Noatak, (c) Seward Peninsula, and (d) Southwest. Several clustered data points in (c) lie outside the CAVM definition of**
**tundra. These points were sampled by BLM and are tundra points.**
Points with fire history also varied by when they were sampled relative to the year of most recent fire and how many
times it had burned from 1940 to 2021. Of the points that were sampled pre-fire, almost all fires occurred within one
decade after sampling. In fact, only eight points fell in the 10-19 years-since-sampling bin (Fig. 5: (a)). Of the points
sampled post-fire, the greatest number of points (5,539 points) was sampled within the second decade since fire,
followed by the third decade and then first decade since fire. Still, there were over one hundred points across five
datasets sampled 30 or more years post-fire (Fig. 5 (c)). For both points sampled before and after the most recent
fire, most points had only one fire occurrence between 1940 and 2021. The number of data points falls exponentially
for points burned more than once. There are, however, points that have up to four years of recorded fire for both
points that were sampled before and after the most recent fire (Fig. 5: (b), (d)).
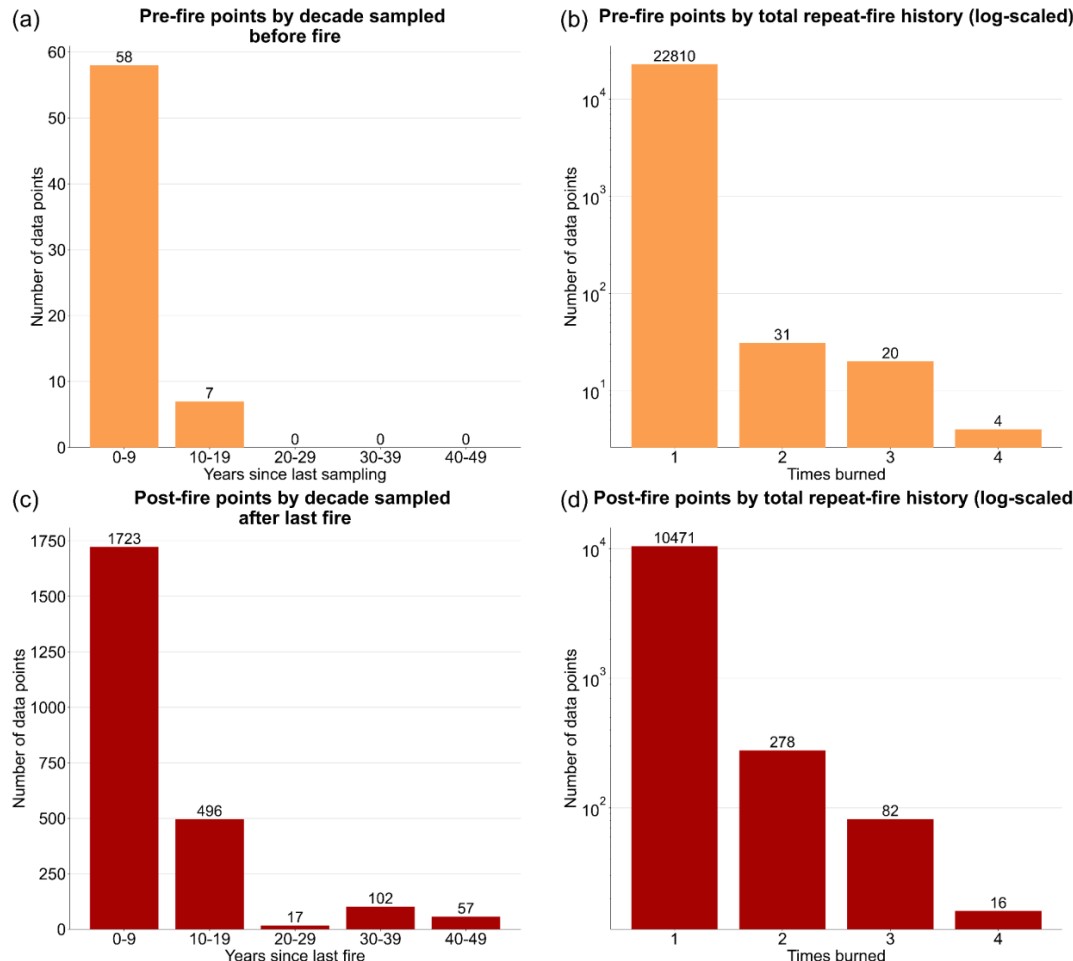

**Figure 5: (a) points sampled before the most recent fire binned by years between sampling and fire disturbance, (b) points sampled before the most recent fire binned by number of times burned, (c) points sampled after the most recent fire binned by years between the last fire and the sampling date, and (d) points sampled after the most recent fire binned by number of times burned.**

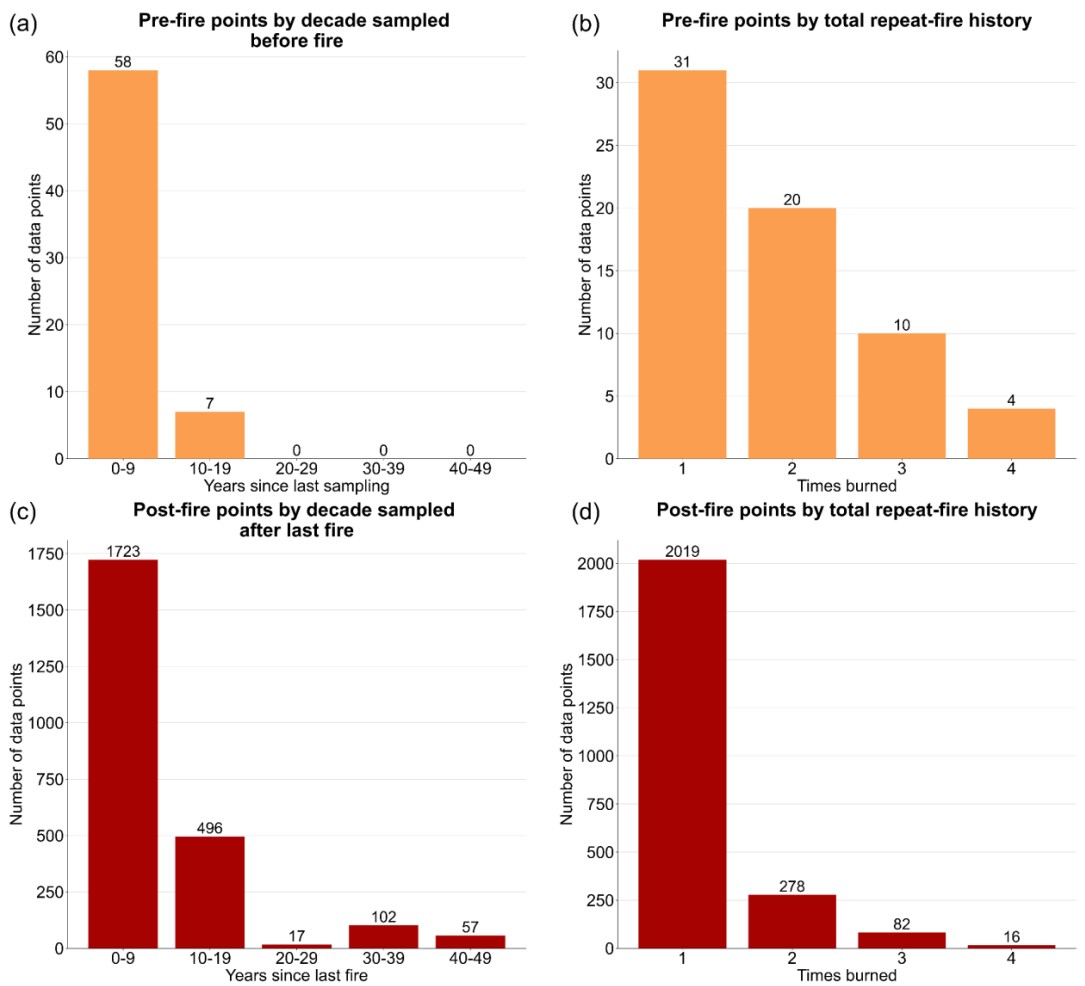

**Figure 6: Excluding the Schaefer_2021 dataset: (a) points sampled before the most recent fire binned by years between sampling and fire disturbance, (b) points sampled before the most recent fire binned by number of times burned, (c) points sampled after the most recent fire binned by years between the last fire and the sampling date, and (d) points sampled after the most recent fire binned by number of times burned.**

Table 4 summarizes datasets within each subregion and their fire history. The greatest number of burned points, both sampled before and after fire appear in Southwest Alaska owing largely to the Schaefer_2021 dataset. The Seward Peninsula subregion, on the other hand, contains the largest number of datasets with fire history. The Noatak subregion has the greatest number of fire years represented in this database with 17 unique fire years, 14 of them included for points within the Loboda_2022 dataset. All fire data from the North Slope, with the exception of some points from a 2017 fire in the Miller_2022 dataset, are from the 2007 Anaktuvuk River Fire (Fig. 4; Table 4).



**Table 4: Fire history for points from the ALFD by subregion and datasets. The dataset name follows the convention of**
**"Name_Year" where "Name" indicates the names of the principal investigators and "Year" is the year of the data release.**
**If the original dataset has not been released publicly, the year of the data acquisition was used.**

| Subregion | Dataset | Burn years* | Number of post-fire points | Number of pre-fires points |
|---|---|---|---|---|
| North Slope | Shaver_2016 | 2007 | 1074 | 0 |
| | Schaefer_2021 | 2007 | 285 | 0 |
| | Rocha_2015 | 2007 | 123 | 0 |
| | Miller_2022 | 2007, 2017 | 76 | 0 |
| | Mack_2011 | 2007 | 22 | 0 |
| | Rocha_2020 | 2007 | 8 | 0 |
| Noatak | Loboda_2022 | 1971, 1972, 1976, 1983, 1984, 1985, 2000, 2002, 2003, 2004, 2005, 2010, 2012, 2014 | 504 | 0 |
| | Jorgenson_2018 | 1972, 1977, 1994, 1999, [2010, 2012] | 16 | 25 |
| Seward Peninsula | Tsuyuzaki_2013 | 2002 | 210 | 0 |
| | Loboda_2022 | 1954, 1971, 1997, 2002, 2015, [2019] | 168 | 19 |
| | Hollingsworth_2020 | 1971, 2002, [2015] | 15 | 5 |
| | Iwahana_2005 | 2002, [2019] | 8 | 8 |
| | Raynolds_2018 | 1971, [2002, 2019] | 4 | 3 |
| | Jandt_1995 | 1957, 1977, [2005] | 3 | 2 |
| | Berner_2018 | [2002, 2015, 2019] | 0 | 3 |
| Southwest | Schaefer_2021 | 1985, 2006, [2015] | 8167 | 22800 |
| | Natali_2022 | 1972, 2015 | 124 | 0 |
| | Frost_2020 | 1971, 1972, 1985, 2006, 2007, 2015 | 40 | 0 |

*Burned points sampled pre-fire appear in square brackets ([])
**5 Discussion**
**5.1 Scientific implications**
SATFiD represents the first effort we know of to compile the field datasets of vegetation, active layer properties,
and fire history collected in different parts of the Alaskan tundra and reconcile them into a consistent database. As
such, it offers the largest collection of Alaskan tundra field data accessible in one place. It spans both a large





temporal extent of 49 years and spatial extent, with over 1,000 data points coming from each of the four subregions
of the Alaskan tundra.
The descriptive analyses provided here provide examples of and a starting point for exploring the database and its
coverage of various variables spatially and temporally. With this rich resource of in-situ measurements, we
encourage future investigators to identify potential research applications and questions that can be asked with this
database. Possibilities may involve relating soil variables and vegetation cover to fire history. Studies could look at
patterns or differences over spatial extents or between different subregions. They might also consider patterns or
trends over time. Researchers could also leverage the database as training points for remote sensing based, spatially
explicit or physical, process-based modeling. Variables such as vegetation cover and soil variables such as soil
moisture, soil temperature, and active layer thickness could potentially feed into these models.
Another benefit and potential use of this synthesized database is in discovering opportunities for future research.
One aspect of field studies in the Alaskan tundra that we found while compiling the database is that revisits and
repeat observations over many years is lacking, likely due in part to the difficulty of accessing the regions where the
initial studies took place and limitations placed by government funding that generally favors short-term (3-4 year)
studies. As the climate, soil, and vegetation features of the tundra transform, it would be opportune to revisit points
in this database in order to measure changes and trends over time. The descriptive analysis we conducted also
indicates that a large number of points were burned in the years after field sampling took place, which we've called
"pre-fire" points (Fig. 3). These points can be examined by subregion (Fig. 4, Table 4), and information on the
number of times burned and how many years passed between the sampling and fire occurrence can be found in the
database (Fig. 5, 6). Selecting and revisiting these points based on this fire history information could form the basis
for studies on pre- and post-fire analysis of change. SATFiD can also inform future research by providing a broad-
scale idea of what variables could be of interest and the common methods used to measure them. This could be a
step leading towards greater standardization in variables measured and the techniques used, which would strengthen
future sampling and synthesis research efforts.
Although there are a large number of points dispersed throughout the four subregions of the Alaskan tundra, the map
of the 197,830 unique data points in SATFiD also demonstrates strong geographic clustering. This makes intuitive
sense as in-situ studies of this remote region are challenging, and investigators typically collect large quantities of
data within their relatively small, accessible study areas. Based on this database, future researchers can also identify
areas that have not been sampled before that may be interesting for ecological reasons and fill gaps in data
availability as well as knowledge of the various conditions in the heterogeneous tundra landscape. There are also
many areas within fire extents defined by the ALFD that have not been sampled by any datasets ingested in this
database and could be the sites for fire-related field studies.
**5.2 Uncertainty**
The datasets ingested in SATFiD originate from a variety of research efforts led by different principal investigators
and span five decades of field sampling. This leads to large variances in both the documentation and methods



employed for sampling. Often, a same or similar variable is measured slightly differently between datasets. These
differences produce uncertainties that can propagate and influence results in unpredictable ways when conducting
synthesis studies with these data and represent an important consideration for any synthesis work.
In order to help identify potential sources of uncertainty that should be factored or acknowledged in research using
these data, we have compiled variables that commonly have methodological differences among datasets as well as
the common measurement methods applied for each (Table 5). Of particular note is how different datasets have
defined their plots. For many soil and vegetation variables, measurement instrumentation varied as did the number
of samples taken. Another important consideration is that soil moisture tends to vary significantly within and across
seasons. One-time measurements are less meaningful than measurements logged over an entire season or number of
years. For vegetation cover data, the accuracy of cover depends on methodology as some are more quantitative
while others are more qualitative. Also, not all the chosen functional types for this synthesis were included by every
dataset. It is unclear whether these functional types did not exist in the study area or if the categorization schema
was different, in which case they could have been grouped in with other functional types. As an example, several
datasets that measured cover did not include moss or litter covers (Table 5).
An expanded version of Table 5 that lists each dataset and summaries of methods for each variable when provided in
the original dataset can be found with the data release on the ORNL DAAC. We would strongly encourage
investigators to refer to this expanded table as well as the original datasets' metadata and associated paper
publications for additional details in methodology. An important next step for synthesis research using our database
is taking this information, conducting meta-analysis, and finding ways to factor in and address uncertainties.
Fire attributes including fire history information sampled from the ALFD as well as dNBR from the Landsat-derived
Burn Scar dNBR dataset (Loboda et al., 2018) are not comprehensive or perfectly accurate. Before 1987, the ALFD
defined large fires as fires at least 1,000 acres in area. Between 1987 and 2015, fires of at least 100 acres were also
included. Since 2015, fires of at least 10 acres have been added (Kasischke et al., 2002; Alaska Large Fire Database |
FRAMES, 2022). Smaller fires are missing from the record especially earlier in the ALFD record, and some fine
scale heterogeneity of burned versus unburned vegetation is also not captured by the fire polygons (Miller et al.,
2023). Fire history attributes for data points are only as accurate as the ALFD. Likewise, the DNBR field is also
only as accurate as the dNBR dataset it was derived from, which only extends from 1985 to 2015 (Loboda et al.,
2018). Points from the early and more recent years of our database's records do not have this attribute even if they
were burned.
**Table 5: Variables with greatest varied sampling methods and several common measurement methods employed.**

| Variable | Common measurement methods |
|---|---|
| LATITUDE, LONGITUDE | Coordinates given may refer to the center, NE corner, or SE corner of the plot depending on the dataset. Datasets from LTER points often only give coordinates at point, not quadrat level. Data have been averaged as appropriate to the point level. |





| | |
|---|---|
| DATE | Most datasets include the year, month, and day of data collection, but there are several for which the date was specified only as far as the month or year. These are formatted YYYYMM00 and YYYY0000 respectively. |
| PH | pH was measured from free water in a soil pit, directly from the soil at various depths, and from soil samples taken to a lab. |
| SOIL_MOIST_% | Instrumentation varied. Campbell Scientific Hydrosense II handheld probes, ground-penetrating radar, DualEM, and TDR 300 were used. |
| ALT_MEAN_CM | Instrumentation varied. Mechanical probing or ground penetrating radar used. |
| LAI_MEAN | Instrumentation varied. SunScan wands, LAI 2000 Plant Canopy Analyzers, and LI-COR 2200 Plant Canopy Analyzers were used. |
| SHRUB_HEIGHT_CM | In most cases, the mean height from multiple measurements was taken, but in a few cases, only the tallest shrub was measured. When only mean vegetation height is available, this is the height provided. |
| MOSS_COVER_%, LICHEN_COVER_%, GRAMINOID_COVER_%, FORB_COVER_%, SHRUB_COVER_%, BARE_COVER_%, LITTER_COVER_% | Not all datasets that measured vegetation cover included each of these plant functional types. Plot sizes and delineations varied greatly. 1 m x 1 m plots, 10 m x 10 m plots, and plots with a specific radius and transects out from the center were most common. Ocular assessment or visual estimates were the most common measurement methods. Hits recorded by a vertically mounted laser using a vegetation point-intercept (VPI) sampling approach was also common. For these, top cover measurements were prioritized over total cover, which includes all vegetation in the vertical path of the laser hit. |


SATFiD strives to be as comprehensive as possible, but we acknowledge there are published and unpublished
datasets referenced in the literature that we may have missed or were unable to obtain for this synthesis effort. Also,
newer field surveys of the Alaskan tundra from 2020 onward are yet to be added to this current collection. In the
future, we hope to build upon this database by ingesting missed and new datasets. Potential future activities might
also include sampling active layer thickness and soil moisture measurements from aerial remote sensing to in-situ
data points by geographic location similarly to how fire history information and dNBR was collected for the current
database. Future improved remote sensing based datasets for fire history and severity may also enable higher spatial
accuracy and temporal consistency for determining each point's fire history and burn severity.
**6 Data availability**
SATFiD (Chen et al., 2023) is available from the Oak Ridge National Laboratory Distributed Active Archive Center
(ORNL DAAC): https://doi.org/10.3334/ORNLDAAC/2177.



## 7 Conclusion

As warming and other climate drivers continue to induce physical and biological changes in the Alaskan tundra, in-situ field measurements of vegetation, active layer, and fire properties are becoming increasingly important as tools to understand and analyze patterns and trends in the region. We synthesized data from the last half-century of tundra field research into a database with utility for synthesis and future research activities of the Alaskan tundra. We reconciled 197,830 individual data points from 37 datasets into a consistent database with 34 variables. Of these 34 variables, eight fire history variables derived from geospatial and remote sensing datasets provide fire information for data points, allowing for scientific analysis relating vegetation and active layer properties to fire attributes. SATFiD is a database investigators can leverage to engage in collaborative synthesis research as well as use to inform aspects of future studies from research questions to study areas and methodologies. This collaborative effort to synthesize tundra field data fits within the scope of the NASA Arctic-Boreal Vulnerability Experiment (ABoVE) Phase 3 goal of combining efforts of multiple research projects to benefit future research. In the context of climate change and its effects on the Alaskan tundra, we hope that this timely synthesis effort will make the data collected over the last five decades more accessible and help inform and guide future research in this region.

## Appendix A

**Table A1: Reference list for all datasets in the SATFiD.**

| Dataset | Citation |
|---|---|
| AKVEG_2022 | Nawrocki, T.W., A.F. Wells, M.J. Macander, E.M. Powers, L.A. Flagstad, A. Droghini, H.A. Gravely, M.A. Steer, G.V. Frost, T.V. Boucher, C.A. Roland, A.E. Miller, D.K. Swanson, and J.K Johanson. 2022. Alaska Vegetation Plots (AKVEG) Database. University of Alaska Anchorage. https://akveg.uaa.alaska.edu |
| Berner_2018 | Berner, L.T., P. Jantz, K.D. Tape, and S.J. Goetz. 2018. ABoVE: Gridded 30-m Aboveground Biomass, Shrub Dominance, North Slope, AK, 2007-2016. ORNL DAAC, Oak Ridge, Tennessee, USA. https://doi.org/10.3334/ORNLDAAC/1565 |
| Breen_2018a | Breen, A.L.. 2018. Arctic Vegetation Plots in Burned and Unburned Tundra, Alaska, 2011-2012. ORNL DAAC, Oak Ridge, Tennessee, USA. https://doi.org/10.3334/ORNLDAAC/1547 |
| Breen_2018b | Breen, A.L. 2018. Arctic Vegetation Plots, Poplars, Arctic and Interior AK and YT, Canada, 2003-2005. ORNL DAAC, Oak Ridge, Tennessee, USA. https://doi.org/10.3334/ORNLDAAC/1376 |
| Davidson_2018 | Davidson, S.J., and D. Zona. 2018. Arctic Vegetation Plots in Flux Tower Footprints, North Slope, Alaska, 2014. ORNL DAAC, Oak Ridge, Tennessee, USA. https://doi.org/10.3334/ORNLDAAC/1546 |
| Ebersole_2018 | Ebersole, J.J. 2018. Arctic Vegetation Plots at Oumalik, AK, 1983-1985. ORNL DAAC, Oak Ridge, Tennessee, USA. https://doi.org/10.3334/ORNLDAAC/1506 |
| Frost_2020 | Frost, G.V., R.A. Loehman, P.R. Nelson, and D.P. Paradis. 2020. ABoVE: Vegetation Composition across Fire History Gradients on the Y-K Delta, Alaska. ORNL DAAC, Oak Ridge, Tennessee, USA. https://doi.org/10.3334/ORNLDAAC/1772 |



| | |
|---|---|
| Hollingsworth_2020 | Hollingsworth, T.N., A. Breen, M.C. Mack, and R.E. Hewitt. 2020. Seward Peninsula post-fire vegetation and soil data from multiple burns occurring from 1971 to 2012: "SPANFire" Study Sites Sampled in July 2012. http://www.lter.uaf.edu/data/data-detail/id/752 |
| Iwahana_2005 | Iwahana, G., K. Harada, M. Uchida, S. Tsuyuzaki, K. Saito, K. Narita, K. Kushida, and L.D. Hinzman. 2016. Geomorphological and geochemistry changes in permafrost after the 2002 tundra wildfire in Kougarok, Seward Peninsula, Alaska. Journal of Geophysical Research: Earth Surface 121:1697-1715. https://doi.org/10.1002/2016JF003921 |
| Jandt_1995 | 1. Jandt, R., K. Joly, C.R. Meyers, and C. Racine. 2008. Slow recovery of lichen on burned caribou winter range in Alaska tundra: Potential influences of climate warming and other disturbance factors. Arctic Antarctic and Alpine Research 40: 89-95. https://doi.org/10.1657/1523-0430(06-122)[jandt]2.0.co;2;<br>2. Jandt, R.R., and C.R. Meyers. 2000. Recovery of lichen in tussock tundra following fire in northwestern Alaska. In: US Department of the Interior, Bureau of Land Management, Alaska State Office. https://doi.org/10.5962/BHL.TITLE.61209 |
| Jorgenson_2018 | Jorgenson, M.T. 2018. Arctic Vegetation Plots in NPS Arctic Network Parks, Alaska, 2002-2008. ORNL DAAC, Oak Ridge, Tennessee, USA. https://doi.org/10.3334/ORNLDAAC/1542 |
| Kade_2018 | Kade, A.N. 2018. Arctic Vegetation Plots at Frost Boil Sites, North Slope, Alaska, 2000-2006. ORNL DAAC, Oak Ridge, Tennessee, USA. https://doi.org/10.3334/ORNLDAAC/1361 |
| Loboda_2022 | Loboda, T.V., L.K. Jenkins, D. Chen, J. He, and A. Baer. 2022. Burned and Unburned Field Site Data, Noatak, Seward, and North Slope, AK, 2016-2018. ORNL DAAC, Oak Ridge, Tennessee, USA. https://doi.org/10.3334/ORNLDAAC/1919 |
| Macander_2021 | Macander, M.J., G.V. Frost, P.R. Nelson, and C.S. Swingley. 2020. ABoVE: Tundra Plant Functional Type Continuous-Cover, North Slope, Alaska, 2010-2015. ORNL DAAC, Oak Ridge, Tennessee, USA. https://doi.org/10.3334/ORNLDAAC/1830 |
| Mack_2011 | Mack, M. 2016. Characterization of burned and unburned moist acidic tundra sites for estimating C and N loss from the 2007 Anaktuvuk River Fire, sampled in 2008. ver 5. Environmental Data Initiative. https://doi.org/10.6073/pasta/81868b65c853d5eb2052d9f1a8397d0d |
| Miller_2022 | Miller, E.A., R. Jandt, C.A. Baughman, B.M. Jones, and D.A. Yokel. 2022. ABoVE: Post-Fire and Unburned Field Site Data, Anaktuvuk River Fire Area, 2008-2017. ORNL DAAC, Oak Ridge, Tennessee, USA. https://doi.org/10.3334/ORNLDAAC/2119 |
| Natali_2022 | 1. Ludwig, S., R.M. Holmes, J. Schade, S. Natali, and P. Mann. 2018. Polaris Project 2017: Vegetation biomass, carbon, and nitrogen, Yukon-Kuskokwim Delta, Alaska. Arctic Data Center. https://doi.org/10.18739/A2FJ29D12;<br>2. Ludwig, S., R.M. Holmes, S. Natali, P. Mann, and J. Schade. 2018. Polaris Project 2017: Soil fluxes, carbon, and nitrogen, Yukon-Kuskokwim Delta, Alaska. Arctic Data Center. https://doi.org/10.18739/A2Q23R08G;<br>3. Natali, S. 2018. Yukon-Kuskokwim Delta fire: thaw depth, soil temperature, and point-intercept vegetation, Yukon-Kuskokwim Delta Alaska, 2015-2016. Arctic Data Center. https://doi.org/10.18739/A2707WP16;<br>4. Ludwig, S., R.M. Holmes, S. Natali, J. Schade, and P. Mann. 2018. Yukon-Kuskokwim Delta fire: vegetation biomass, Yukon-Kuskokwim Delta Alaska, 2016. Arctic Data Center. https://doi.org/10.18739/A29S1KK6T;<br>5. Olefeldt, D., M. Hovemyr, M. Kuhn, D. Bastviken, and T. Bohn. 2021. The fractional land cover estimates from the Boreal-Arctic Wetland and Lake Dataset (BAWLD), 2021. Arctic |



Data Center. https://doi.org/10.18739/A2C824F9X

Raynolds_2018    Raynolds, M.K. 2018. Arctic Vegetation Plots ATLAS Project North Slope and Seward Peninsula, AK, 1998-2000. ORNL DAAC, Oak Ridge, Tennessee, USA. https://doi.org/10.3334/ORNLDAAC/1541

Rocha_2015    Rocha, A., and G. Shaver. 2016. Anaktuvuk River fire scar thaw depth measurements during the 2008 to 2014 growing season ver 6. Environmental Data Initiative. https://doi.org/10.6073/pasta/93121fc86e6fbcf88de4a9350609aed6

Rocha_2020    Rocha, A. 2020. Leaf area index (LAI) recorded from a nitrogen (N), phosphorus (P) and N+P fertilization experiment at the 2007 Anaktuvuk River, Alaska, USA fire scar during the 2016-2019 growing seasons ver 2. Environmental Data Initiative. https://doi.org/10.6073/pasta/06559231aa04fd7fecd661f107985c8f

Schaefer_2021    Schaefer, K., L.K. Clayton, M.J. Battaglia, L.L. Bourgeau-Chavez, R.H. Chen, A.C. Chen, J. Chen, K. Bakian-Dogaheh, T.A. Douglas, S.E. Grelick, G. Iwahana, E. Jafarov, L. Liu, S. Ludwig, R.J. Michaelides, M. Moghaddam, S. Natali, S.K. Panda, A.D. Parsekian, A.V. Rocha, S.R. Schaefer, T.D. Sullivan, A. Tabatabaeenejad, K. Wang, C.J. Wilson, H.A. Zebker, T. Zhang, and Y. Zhao. 2021. ABoVE: Soil Moisture and Active Layer Thickness in Alaska and NWT, Canada, 2008-2020. ORNL DAAC, Oak Ridge, Tennessee, USA. https://doi.org/10.3334/ORNLDAAC/1903

Schickhoff_2018    Schickhoff, U. 2018. Arctic Vegetation Plots in Willow Communities, North Slope, Alaska, 1997. ORNL DAAC, Oak Ridge, Tennessee, USA. https://doi.org/10.3334/ORNLDAAC/1368

Shaver_2012a    Shaver, G. 2012. Leaf Area Index every 15 cm of 1m x 1m chamber flux and point frame plots and sites where dataloggers monitored PAR above, within and below S. pulchra and B. nana canopies during the growing season at the Toolik Field Station in AK, Summer 2012. Environmental Data Initiative. https://doi.org/10.6073/pasta/627698983259d6963a6083d5251723cc

Shaver_2012b    Shaver, G. 2023. Summary of three different Leaf Area Index (LAI) methodologies of 19 1m x 1m point frame plots sampled near the LTER Shrub plots at Toolik Field Station in AK the summer of 2012. Environmental Data Initiative. https://doi.org/10.6073/pasta/17302da4bd951a9dc4140187f03fae24

Shaver_2013    Shaver, G. 2013. Summary of soil temperature, moisture, and thaw depth for 14 chamber flux measurements sampled near LTER shrub sites at Toolik Field Station, Alaska, summer 2012. Environmental Data Initiative. https://doi.org/10.6073/pasta/7ccf390e6fe4824e93b7a2b844605a40

Shaver_2016    Shaver, G., and J. Laundre. 2016. Summer soil temperature and moisture at the Anaktuvuk River Severely burned site from 2010 to 2013, ver 2. Environmental Data Initiative. https://doi.org/10.6073/pasta/3094e3e293703580c95e17ddce51af65

Sloan_2018    Sloan, V.L. 2018. Arctic Vegetation Plots for NGEE-Arctic at Barrow, Alaska, 2012. ORNL DAAC, Oak Ridge, Tennessee, USA. https://doi.org/10.3334/ORNLDAAC/1505

Tsuyuzaki_2013    Tsuyuzaki, S., Iwahana, G., & Saito, K. (2018). Tundra fire alters vegetation patterns more than the resultant thermokarst. Polar Biology, 41, 753-761. https://doi.org/10.1007/s00300-017-2236-7

Tweedie_2018    Tweedie, C.E., P.J. Webber, V. Komarkova, and S. Villarreal. 2018. Arctic Vegetation Plots at Atqasuk, Alaska, 1975, 2000, and 2010. ORNL DAAC, Oak Ridge, Tennessee, USA. https://doi.org/10.3334/ORNLDAAC/1371





| | |
|---|---|
| Walker_2018a | Walker, D.A. 2018. Arctic Vegetation Plots Legacy Project Barter Island and Point Barrow, Alaska, 1994. ORNL DAAC, Oak Ridge, Tennessee, USA. https://doi.org/10.3334/ORNLDAAC/1534 |
| Walker_2018b | Walker, D.A. 2018. Arctic Vegetation Plots, Prudhoe Bay ArcSEES Road Study, Lake Colleen, Alaska, 2014. ORNL DAAC, Oak Ridge, Tennessee, USA. https://doi.org/10.3334/ORNLDAAC/1555 |
| Walker_2018c | Walker, M.D. 2018. Arctic Vegetation Plots from Pingo Communities, North Slope, Alaska, 1984-1986. ORNL DAAC, Oak Ridge, Tennessee, USA. https://doi.org/10.3334/ORNLDAAC/1507 |
| Walker_2018d | Walker, D.A. 2018. Arctic Vegetation Plots at Happy Valley, Alaska, 1994. ORNL DAAC, Oak Ridge, Tennessee, USA. https://doi.org/10.3334/ORNLDAAC/1354 |
| Walker_2018e | Walker, D.A. 2018. Arctic Vegetation Plots at Imnavait Creek, Alaska, 1984-1985. ORNL DAAC, Oak Ridge, Tennessee, USA. https://doi.org/10.3334/ORNLDAAC/1356 |
| Walker_2018f | Walker, D.A. 2018. Arctic Vegetation Plots at Toolik Lake, Alaska, 1989. ORNL DAAC, Oak Ridge, Tennessee, USA. https://doi.org/10.3334/ORNLDAAC/1333 |
| Webber_2018 | Webber, P.J., S. Villarreal, and C.E. Tweedie. 2018. Arctic Vegetation Plots for IBP Tundra Biome, Barrow, Alaska, 1972-2010. ORNL DAAC, Oak Ridge, Tennessee, USA. https://doi.org/10.3334/ORNLDAAC/1535 |
| Williams_1999 | Williams, M., and E. Rastetter. 1999. Measurements of Leaf area, foliar C and N for 14 sites along a transect down the Kuparuk River basin, summer 1997, North Slope, Alaska. Environmental Data Initiative. https://doi.org/10.6073/pasta/a5a4d4154e0a8181a5523b4d9c49ed99 |




**Earth System** Discussions
**Science**
**Data**

**Appendix B**

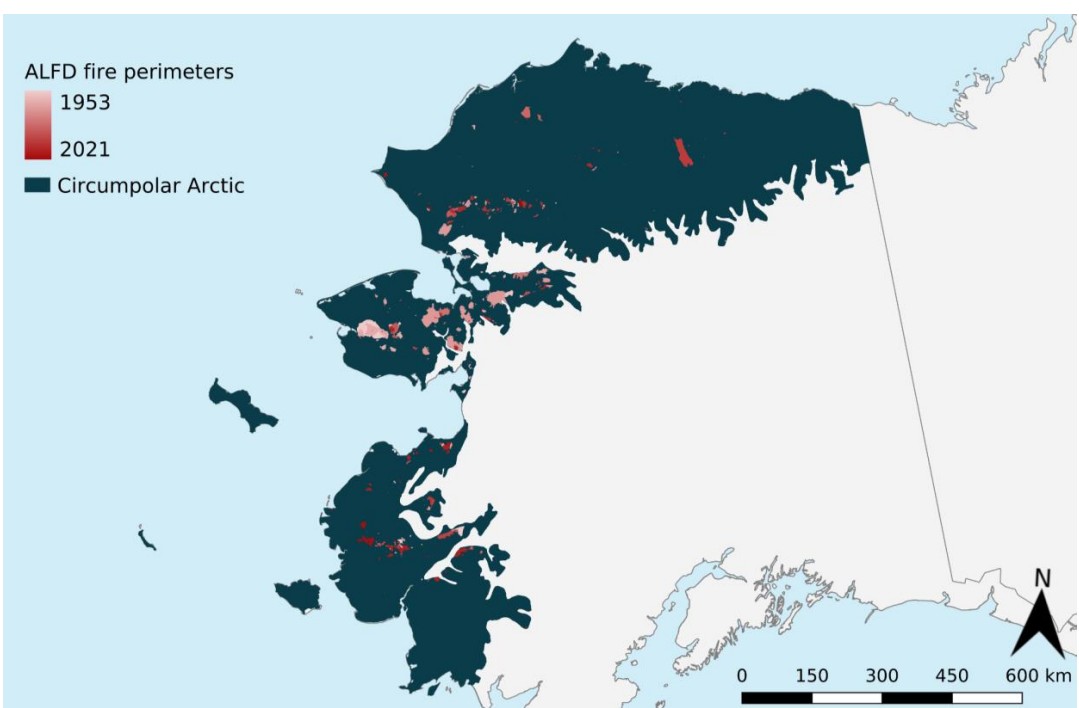

**Figure B1: Map of the Alaska Large Fire Database (ALFD) circumpolar Arctic fire perimeters through 2021.**
**Author contributions**
DC designed the synthesis project. DC and MK initiated the process for listing datasets. XZ and DC compiled the
database and wrote the draft. EH mentored XZ and contributed to compiling the database and writing. All authors
contributed to discussing the results and editing of the final paper.
**Competing interests**
The authors declare that they have no conflict of interest.
**Acknowledgments**
This paper was supported by the NASA Arctic-Boreal Vulnerability Experiment (ABoVE) through NASA
Terrestrial Ecology program grants NNX15AT79A and NNH16CP09C; the NASA summer internship program
through the NASA Terrestrial Ecology program and the Carbon Cycle and Ecosystems Office; the College of
Behavioral and Social Sciences at the University of Maryland, College Park through the Dean's Research Initiative
award; NSF-1915307; NSF-2103539; and Gordon & Betty Moore Foundation-#8414. Resources supporting this
work were provided by the NASA High-End Computing (HEC) Program through the NASA Center for Climate
Simulation (NCCS) at Goddard Space Flight Center.



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
