# Peer review of "A synthesized field survey database of vegetation and active"

_Earth System Science Data, 2023_

## Author Response (AR1)

Dear Referees,

Thank you for your reviews and encouraging, insightful comments for our manuscript. We greatly appreciate that you see the value in this data synthesis effort and are grateful for your helpful feedback. We have incorporated your suggestions into our revised manuscript. Please also find our responses (*in italics*) to each suggestion and corresponding changes (**in bold**) in the manuscript bulleted below.

**RC1:** The only thing I would like to see added is a description of whether or not this database will be open to new submissions in the future and if so, how it will be maintained.

- *We agree that we should include a description of whether and how we plan to maintain this database. We intend to update SATFiD biennially to include newly acquired field data in the Alaskan tundra, allowing the further expansion of SATFiD's utility in studies of long-term changes in the tundra. We will actively seek funding to support these future updates. This information has been added to section 5.1 Scientific Implications within the Discussion section.*
- 5.1 Scientific Implications, Lines 323-325: **Additionally, we intend to keep SATFiD updated biennially to include newly acquired field data in the Alaskan tundra, allowing the further expansion of SATFiD's utility in studies of long-term changes in the tundra. To that end, we will actively seek funding to support the future updates.**

**RC2:** I suggest to add a fourth one [objective]: to provide basic data on properties of the components of the ecosystem such as soil physical properties, active layer thickness, organic layers, water content, vegetation composition and dynamics, as measured parameters for modelers to validate their simulations as well as for providing a better understanding of processes for their modeling.

- *Thank you for this suggestion. It is true that this database can function as a resource for modelers, and we have added this in as a fourth objective in the Introduction.*
- 1 Introduction, Lines 103-119: Because of this, we built a database from *in situ* datasets across the Alaskan tundra with **four** major objectives: (1) Gather datasets and synthesize them in a way that will facilitate further analysis by investigators and promote synthesis research efforts, (2) deepen our understanding of ecosystem processes within the Alaskan tundra, particularly fire-vegetation-permafrost interactions,  (3) identify areas of interest for future research where knowledge is lacking or there is great potential for follow-up research to study change and long-term trends**, and (4) provide a source of vegetation and soil properties data that could improve understanding of physical processes and be used to inform and validate process-based models and simulations**.

**RC2:** The authors are conscious that the data compiled in the different regions of the Alaskan tundra provide "snapshots" of the state of the vegetation and basic soil conditions at the time the various surveys were carried on the field. Maybe, this should be explained in a stronger statement in the text. [...] Therefore, although providing fundamental basic information, the SATFiD data will need to be used with some circumspection.

- *As you indicate, we are well aware of the limitations of the field data and hope to convey the importance of considering them when using the data. To clarify this, we have added a statement at the end of section 5.2 Uncertainties to convey that users should take into account this non-static nature of the Arctic tundra when adopting SATFiD for long-term analyses, given the various warming-induced ecological changes in the Arctic that are not necessarily captured by the "snapshot" data within this database.*
- 5.2 Uncertainties, Lines 368-373: **One additional caveat when using SATFiD is its long-term nature. Because it ingests various datasets that were collected over half a century, during which the Arctic tundra has undergone substantial warming (Kaufman et al., 2009), the tundra conditions from the earlier field campaigns may be quite different from those acquired in recent years. For example, two data entries in SATFiD collected decades apart with similar values of certain measurements do not necessarily mean that the two tundra sites that they represent are ecologically similar. Users should take into account this non-static nature of the Arctic tundra when adopting SATFiD for long-term analyses.**

**RC2:** Editing/technical comments
- Line 36: "exceptionally rapid rates of warming", exceptional relative to what? Maybe just say very rapid.
  - *We have rephrased it as "rapid rates of warming".*
  - Abstract, Lines 35-36: Studies in recent decades show strong evidence of physical and biological changes in the Arctic tundra largely in response to  rapid rates of warming.

- Line 54: I suggest replacing "biological changes" by ecological changes.
  - *We have rephrased this as "ecological changes"*
  - 1 Introduction, Lines 53-54: Over recent decades, the Arctic tundra has warmed three to four times faster than the global average (Rantanen et al., 2022), leading to profound physical and  **ecological** changes.

- Line 73: I suggest replacing "strong warming" by fast or rapid warming
  - *We have rephrased it as "rapid warming".*

- ○ 1 Introduction, Lines 73-74: Considering the Arctic tundra's important role in the Earth system and the  **rapid** warming in this region, understanding current ecosystem dynamics is crucial for the projection of future states of the Arctic tundra.

- Line 80: replace "expeditions" by field work or field surveys
  - ○ *We have rephrased it as "field surveys".*
  - ○ 1 Introduction, Lines 78-80: This historical lack of studies compared with other biomes is the consequence of limited *in situ* measurements, stemming from interwoven factors including harsh Arctic environmental conditions, logistical challenges, and the high cost of conducting scientific  **field surveys**.

- Line 81: "The Alaskan tundra is an important component of the Arctic tundra biome…" Maybe say differently like.. represents an important fraction of the Arctic tundra with an area over 8.5 million km2 and shares similarities with other Arctic regions.
  - ○ *We have rephrased this to "The Alaskan tundra represents an important fraction of the Arctic tundra biome that spans over 8.5 million km² and shares similar characteristics with other Arctic regions".*
  - ○ 1 Introduction, Lines 81-82: The Alaskan tundra  **represents** an important  **fraction** of the Arctic tundra biome that spans over 8.5 million km² and  **shares similar characteristics with other Arctic regions** (CAVM Team, 2003).

- Line 86: I suggest replace "included" by inventoried.
  - ○ *We have rephrased it as "inventoried".*
  - ○ 1 Introduction, Lines 85-87: However, even these early records of wildfires across the region are sparse, and often only larger wildfires were  **inventoried**, leading to unaccounted wildfires in the region (Miller et al., 2023).

- Line 97: delete "we argue that", not necessary.
  - ○ *Addressed.*
  - ○ 1 Introduction, Lines 96-99: While all existing field datasets are important in their own right (in support of the scientific goals of the individual field campaigns),  when combined properly they can provide an unprecedented lens through which the ecosystem dynamics of the Arctic tundra, both aboveground and below-ground, can be revealed at a wide spatial scale.

- Table 2: last procedure: daily calculation. Unclear to me. Do you mean repeat measurements over a long time period, i.e. months, years?? As it is written now I understand that the mean of a small number of measurements taken within one day is

considered a daily mean..?!. Maybe I misunderstand… For instances, are temperature data taken over time with dataloggers averaged daily?

- ○ *Your understanding of the data measured at the same site being averaged daily is correct. This is done in cases when multiple samples were taken for certain variables. Average values are reported in this synthesis. We rephrased this in the chart as "Repeat measurements from the same day and plot, as defined by the latitude and longitude, were averaged for all quantitative variables."*
- ○ 3.2 In-situ variables selection, Line 171/Table 2:

| Daily mean calculation | Repeat measurements  from  **the same day and** plot, as defined by the latitude and longitude,  were averaged for all quantitative variables. |
| --- | --- |

- ● Figures 5 and 6. I suggest you reorganize the figures and captions so as not to repeat a and c in both figures.
  - ○ *Thank you for pointing this out. In Figure 5, a and c should be the distribution of* all *burned data points within our dataset by fire history, whereas in Figure 6, a and c exclude the Schaefer_2021 dataset as it dominates the data. We have updated Figure 5 to show the correct plots a and c. On the other hand, we will remove Figure 6 as it is repetative, serving only to show the data distribution when taking Schaefer_2021 data out. We already discuss the dominance of this data at various points in the text as well as in Figures 2 & 3 and Table 4. As Figure 6 was not referenced in the text and does not contribute sufficient new, meaningful information, we will exclude it from the revised manuscript.*
  - ○ 4.2 Descriptive analysis of data by fire attributes, Line 262/Figure 5 (updated):